# Successful Treatment of SAPHO Syndrome Complicated with Ankylosing Spondylitis by Secukinumab: A Case Report

**DOI:** 10.3390/jpm13030516

**Published:** 2023-03-13

**Authors:** Wei Tu, Daan Nie, Yuxue Chen, Cheng Wen, Zhipeng Zeng

**Affiliations:** 1Department of Rheumatology and Immunology, Tongji Hospital, Tongji Medical College, Huazhong University of Science and Technology, Wuhan 430030, China; 2Department of Cardiology, Union Hospital, Tongji Medical College, Huazhong University of Science and Technology, Wuhan 430022, China; 3Department of Rheumatology and Immunology, Xiaogan First People’s Hospital, Xiaogan 432000, China

**Keywords:** SAPHO syndrome, ankylosing spondylitis, secukinumab

## Abstract

Synovitis, acne, pustulosis, hyperostosis, and osteitis (SAPHO) syndrome is characterized by a wide range of dermatological and musculoskeletal manifestations, and its outcome has recently been improved greatly by optimizing management. However, the treatment strategies are not standardized and require further refinement. Secukinumab, a fully human monoclonal antibody targeting IL-17A, is approved for the treatment of autoimmune psoriasis, psoriatic arthritis (PsA), and ankylosing spondylitis (AS). Here, a 53-year-old man was diagnosed with AS, and he presented scattered pustulosis in both hands and feet with a 5-year history of recurrent lumbosacral area pain and abnormal pain in the neck and front chest area. Secukinumab improved the patient’s cutaneous lesion and prevented musculoskeletal pain by substituting adalimumab. Although only a few cases have been reported that secukinumab can effectively treat SAPHO syndrome complicated with AS, the efficacy remains controversial. Therefore, we hope to provide a novel valuable therapeutic strategy for SAPHO syndrome management, particularly in patients with skin lesions.

## 1. Background

Synovitis, acne, pustulosis, hyperostosis, and osteitis (SAPHO) syndrome, first reported by Chamot in 1987, is a rare type of chronic inflammatory disease characterized by dermatological and musculoskeletal manifestations with a prevalence of less than 1/10,000 [1,2]. The canonical characteristics of SAPHO syndrome consist of cutaneous lesions and osteoarticular entanglement [3]. The cutaneous lesions generally present as palmoplantar pustulosis (PPP), severe acne, and psoriatic skin lesions [3]. Osteoarticular involvement mainly includes osteoarthritis, bone hypertrophy, synovitis, arthritis, and enthesitis. The primary affected location is the anterior chest wall (mainly including sternoclavicular joint, sternocostal joint, and sternal handle sternal joint), followed by the spine and sacroiliac joint [2,3]. Ankylosing spondylitis (AS), the prototype of the spondyloarthropathies (SpA), is a common chronic inflammatory disease that is mainly characterized by pain in the back, gluteal region, or lumbosacral area, largely due to inflammation of the spine and sacroiliac joints [4]. However, there is currently no effective approach for SAPHO management [3,5]. Secukinumab is a fully human monoclonal antibody targeting IL-17A which has been approved for the treatment of autoimmune psoriasis, psoriatic arthritis (PsA), and AS. Recently, secukinomab has been emerging as a promising treatment strategy for refractory SAPHO syndrome [6,7]. In the present case, we reported an extremely rare case of SAPHO syndrome complicated with AS with successful and complete resolution after treatment with secukinumab.

## 2. Case Presentation

A 53-year-old man who presented with a 5-year history of recurrent lumbosacral area pain, accompanied by pain in neck and front chest area, was admitted to our hospital in March 2021. Physical examination showed that pustulosis was scattered in both hands and feet (Figure 1), accompanied by tenderness of the sternum. The Schober test was negative. Patrick signs were positive on both sides. 

Laboratory tests results showed that human leukocyte antigen B27 (HLA-B27) was positive, hypersensitive C-reactive protein (hs-CRP) was 46.69 mg/L, and erythrocyte sedimentation rate (ESR) was 49 mm/h. A computed tomography (CT) showed multiple sites of wormhole-like bone destruction on the bilateral sacroiliac joint surface, and the right sacroiliac joint was partially fused, in accordance with grade 2 to 3 bilateral sacroiliitis (Figure 2A); a magnetic resonance image (MRI) showed inflammation in the sacroiliac joint (Figure 2B). On the basis of the above results, the diagnosis of AS was determined, and the manifestations met the modified New York criteria in 1984.

The skin lesions were diagnosed with palmoplantar pustulosis (PPP) after consultation with dermatologists. Notably, a chest CT image showed obviously enlarged sternoclavicular joints with marked hyperostosis and osteosclerosis (Figure 2C), as indicated by the characteristic “bull’s head sign” [8]. A ^99^Tc^m^-methylene diphosphonic acid (^99^Tc^m^ -MDP) single-photon emission computed tomography (SPECT) whole-body bone scan revealed a remarkable radioactive accumulation in the cervical vertebra, sternoclavicular joint, first anterior rib, sternum, and bilateral sacroiliac joints (Figure 3). Thereby, the diagnosis of SAPHO syndrome could be established on the basis of comprehensive clinical assessment.

Then, the patient was treated as follows: diclofenac sodium dual release eneric-coated capsule (75 mg po bi-diurnally) for 1 month, and adalimumab injections (40 mg s.c., every 2 weeks) for 3 months. The above treatment effectively relieved the patient’s lumbosacral area pain and chest pain, as well as the pustulosis. However, extensive erythema and flaking skin on hands and feet appeared, covered with dry scales, after 2 months of treatment (Figure 4A). After evaluation by dermatologists, adalimumab (40 mg s.c., every 2 weeks) was discontinued, and subcutaneous secukinumab was administrated at 150 mg weekly for the first month and monthly thereafter, while compound flumetasone was provided for external use. Two months later, the patient achieved complete remission of skin lesions in the palms and feet (Figure 4B), with almost no joint pain and no adverse reaction, as indicated by the laboratory tests results of ESR (3 mm/H) and hs-CRP (1.5 mg/L). The patient then returned to the hospital every month to report any potential adverse reactions. As of 6 January 2023, his symptoms had resolved completely without any complications or adverse reactions. 

## 3. Discussion 

SAPHO syndrome is a rare disease without definite etiology [3,5]. The epidemiological data from China show that the syndrome is mainly prevalent in the population aged 30–50, with a male-to-female ratio of 1:2 [2]. Although the prevalence is less than 1 in 10,000 in Caucasians, some scholars have suggested that its real morbidity may be substantially underestimated [2]. However, there is a consensus that SAPHO syndrome may occur at any age with an underestimated incidence from 15-month-old infants to 79-year-old seniors [2,9]. The elders at onset may display a spectrum of symptoms consisting of bone and joint (axial and peripheral bone and joint manifestations), but middle-aged patients are predominantly prevalent, with PPP accompanying cutaneous lesions, while young patients usually present with severe acne [9]. The present case was consistent with this feature, accompanied by both osteoarticular involvement and cutaneous lesions. 

The most frequent referential diagnostic criteria for SAPHO syndrome were proposed by Kahn and Khan in 1994, and revised by Kahn in 2003 [10], including bone–joint involvement associated with isolated palmoplantar pustulosis (PPP), and bone–joint involvement associated with severe acne, isolated or multifocal sterile hyperostosis/osteitis (adults), and chronic recurrent multifocal osteomyelitis (children). The diagnosis must rule out septic ostemomyelitis, bone neoplasms, and osteoarthritis. The pathogenesis of SAPHO syndrome remains elusive; however, some specialists have proposed that bacterial infection and immunologic dysfunction contribute to the development and progress of SAPHO syndrome, as in the case of *Propionibacterium acnes* infection [3]. Anaerobic microorganisms may contribute to the development of SAPHO syndrome by inducing T-lymphocyte-mediated humoral immunity and inflammatory responses. The relationship between tonsillitis and SAPHO syndrome has gradually emerged as a promising research area. Tonsils are a common site for most insidious chronic infections. Recently, a study reported that SAPHO syndrome with tonsillitis accounted for approximately two-thirds of SAPHO patients, and those cases presented more serious skin and nail lesions than cases without tonsillitis [11]. Tonsillectomy may contribute to the improvement of bone and skin symptoms in SAPHO patients [11]. *Staphylococcus aureus* and *Propionibacterium acnes* are the most common pathogenic microorganisms causing recurrent tonsillitis, which have been isolated from the bone biopsies of patients with SAPHO syndrome [12]. Hence, tonsillitis is deeply associated with the development and progression of the disease. However, it has also been suggested that SAPHO syndrome belongs to the category of SpA, and that the inflammatory osteoarthritic symptoms may be a trigger factor for disease development [5,13]. SAPHO syndrome is associated with psoriasis, PsA, and AS, and it may have a common pathogenesis. Unlike the high positive rate of HLA-B27 in SpA, only 2.4% to 13% of cases are HLA-B27 positive in SAPHO syndrome [2,14]. Some proinflammatory cytokines, including serum tumor necrosis factor (TNF)-α, interleukin (IL)-1, IL-8, and IL-17, as well as Th17 cells, are elevated in SAPHO patients [5]. These data indicate that immunologic dysfunction plays an essential role in the progression of SAPHO syndrome. 

In this case, the patient presented typical manifestations of SAPHO syndrome: pain in the anterior chest wall, bone inflammation on SPECT of whole-body bone scanning, “bull’s head sign” on chest CT, PPP on the skin, and plaque psoriasis. After ruling out bone neoplasms and infectious inflammation, the patient was diagnosed with SAPHO syndrome according to the diagnostic criteria for SAPHO syndrome proposed by Kahn and Khan [10]. SAPHO syndrome is prone to misdiagnosis for a variety of atypical manifestations, and due to confusion with many diseases including AS, bone tuberculosis, and bone neoplasms. The primary dilemma is that the first symptoms vary among patients with SAPHO syndrome that hinder differential diagnosis. Furthermore, the lack of awareness of manifestations among clinicians can also lead to the underestimated diagnosis of SAPHO syndrome. Therefore, if patients are suspicious of SAPHO syndrome, they are recommended for local image examination and a whole-body bone scan to detect potential lesions and substantiate definite diagnosis.

At present, although there are no specific markers for SAPHO syndrome evaluation, ESR and CRP may increase during the active phase [15]. In the present case, ESR and hs-CRP were significantly elevated at admission, which indicated the active stage of SAPHO syndrome; levels were reduced to normal levels after treatment, suggesting remission of the disease. Therefore, it is worth monitoring the changes in ESR and hs-CRP to evaluate the progression and guide the management of SAPHO syndrome [15]. Image examination is essential for the diagnosis, evaluation, and treatment of SAPHO syndrome [5,16]. A ^99^Tc^m^-MDP whole-body bone scintigraphy is highly appreciated as the preferred image test in the diagnosis of SAPHO syndrome for its high sensitivity in detecting the involvement of bones and joints [16]. In particular, when the sternoclavicular joint, the first sternocostal joint, and the sternocostal–shank junction are involved simultaneously in patients with SAPHO syndrome, the typical “bull’s head sign” emerges, characterized by enhanced radioactive accumulation [15]. However, only approximately 20% of patients with SAPHO syndrome manifest the “bull’s head sign” [17]. Additionally, the patient showed hyperplastic hypertrophy lesion at the sternoclavicular joint on chest CT, and he was unequivocally diagnosed with SAPHO syndrome. Noticeably, image tests are necessary for the diagnosis and evaluation of SAPHO syndrome, and they should be recommended in all suspicious patients.

AS is a chronic inflammatory disease featured by inflammation in sacroiliac joints and spinal attachment points, and is strongly associated with HLA-B27 [4]. Peripheral arthritis, acute anterior uveitis, inflammatory bowel disease (IBD), and psoriasis are frequently present in AS [18]. At an early phase, the sacroiliac joint is firstly influenced as the presumed initial manifestation in the AS patient [4,18]. A whole-body bone imaging can provide a sensitive visualization of inflammatory lesions in the sacroiliac joint and contribute to the early diagnosis and prognosis evaluation of AS [19]. In addition, CT and MRI, which are sensitive alternatives for AS diagnosis and evaluation, can detect potential lesions in the spine, sacroiliac joint, and peripheral joints in SAPHO syndrome. However, despite its superiority in evaluating the involvement of the whole body and joints, PET-CT is not available as a major scan tool because of the high cost [20]. Therefore, the selection of image tests should depend on the specific condition of patients with SAPHO syndrome.

The primary purpose for SAPHO syndrome management is to relieve the symptoms, including alleviating pain and ameliorating skin lesions [3,21]. Nonsteroidal anti-inflammatory drugs (NSAIDs) are first-line drugs for relieving pain but not for preventing progress in SAPHO syndrome [3,21]. However, a variety of regimens, including glucocorticoids, bisphosphonates, some traditional antirheumatic drugs (such as methotrexate and leflunomide), and traditional Chinese medicines, have been reported to be effective in curbing disease deterioration and improving symptoms in SAPHO patients [21]. Both biological agents and JAK inhibitors have also achieved satisfactory remission, especially in patients with significantly elevated inflammatory levels after ineffective conventional drug treatment [6,7,11]. Since *Propionibacterium acnes* is a potential trigger for SAPHO syndrome, antibiotics may be an efficacious option for SAPHO syndrome treatment [22]. It has been proven that the acne in the patients with SAPHO syndrome is sensitively responsive to antibiotics, including tetracycline, azithromycin, clindamycin, and cefcapene [12,22,23]. Due to a shortage of sound evidence in the fields of clinical randomized controlled trials, it is difficult to reach a consensus about the treatment of SAPHO syndrome unless a conclusive explication of clinical investigations is elaborated. However, the emergence of new drugs has provided more options for SAPHO management, greatly promoting the remission of the disease, especially in refractory SAPHO syndrome.

Specifically, this case was complicated with AS. To date, only a few cases of SAPHO syndrome with AS have been reported, and the case’s skin lesions and osteoarticular pain were significantly improved by tofacitinib or secukinumab when adalimumab was discontinued [24,25]. Some scholars believe that SAPHO syndrome can be classified in the category of SpA, which is between AS and PsA [3,26]. The patient’s pain and PPP were relieved by adalimumab. However, psoriatic skin lesions on feet and hands appeared after treatment. There are cases of induction or exacerbation of psoriasis in patients with AS, rheumatoid arthritis, Crohn’s disease, and SAPHO syndrome, after receiving therapy with TNF-α inhibitors [27,28]. TNF-α inhibitors have less of an effect on improving skin lesions than on musculoskeletal manifestation, and they may even aggravate skin rash [28]. It was reported that 17% of SAPHO syndrome patients treated with TNF-α antagonists developed secondary paradoxical psoriatic lesions, and three patients showed aggravated palmoplantar pustules [29].

On the basis of a literature review and clinical practice, we decided to replace adalimumab with secukinumab for the SAPHO patient’s management. Secukinumab is a fully human IgG1 anti-IL-17A monoclonal antibody which can selectively inhibit the inflammatory cascade induced by IL-17A. The head-to-head trial showed that secukinumab was superior to adalimumab for treating active PsA with satisfactory success in reliving robust skin outcomes [30]. The European League against Rheumatism (EULAR) also recommends that IL-17 inhibitors may be preferred in patients with significant psoriasis in the 2022 updated recommendations of axial spondyloarthritis [31]. In particular, secukinumab has excellent therapeutic efficacy in patients with SAPHO syndrome, especially for patients with PPP and severe skin lesions [10]. After receiving secukinumab treatment, the patient’s skin lesions improved rapidly within 1 month [10]. Notably, the patient’s back pain improved greatly without relapse after discontinuation of adalimumab. Therefore, it is indisputable that secukinumab can also ameliorate AS during SAPHO syndrome treatment without severe adverse events, and it may tackle any underlying common pathogenesis in AS and SAPHO syndromes.

Briefly, we delineated a case of SAPHO syndrome complicated by AS that attained complete resolution after secukinumab substitution. Therefore, secukinumab could be a preferable optimal treatment for SAPHO syndrome. There were several elements that contributed to the patient’s salvation. Firstly, the patient had more severe bone and joint symptoms than skin lesions when admitted to hospital; thus, adalimumab, which has an obvious therapeutic effect on AS, achieved an excellent curative effect in the patient. Moreover, since skin lesions subsequently became the main symptom, adalimumab was replaced with secukinumab, which has a significant advantage in the treatment of psoriatic lesions.

Therefore, secukinumab is a promising therapeutic approach in SAPHO syndrome complicated with AS. However, we advocate for careful scrutinization of secukinumab intervention on a case-by-case basis. In summary, this report is informative in providing a salutogenic treatment option and paving a novel avenue for delving into potential pathogenesis in patients with SAPHO syndrome.

## Figures and Tables

**Figure 1 jpm-13-00516-f001:**
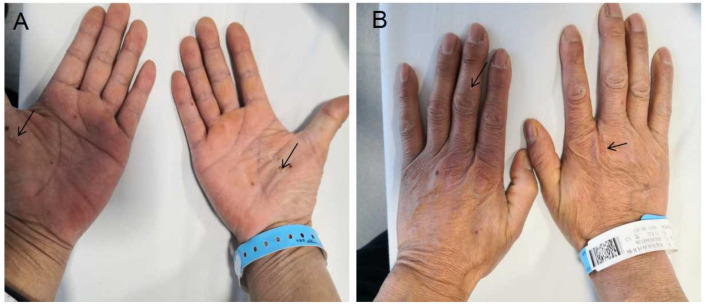
Palmoplantar pustulosis (PPP) on the patient’s palms and feet before treatment. The arrows indicate the palmar pustulosis. (**A**) Palm; (**B**) opisthenar.

**Figure 2 jpm-13-00516-f002:**
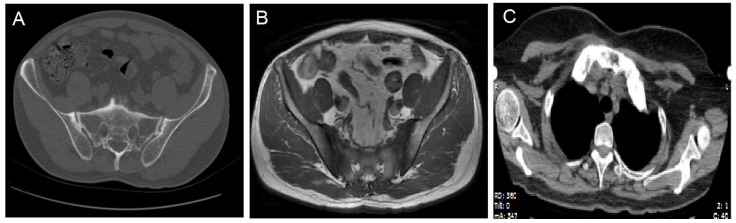
A Computed tomography (CT) image showing multiple sites of wormhole-like bone destruction and partial fusion of right sacroiliac joint (**A**). A magnetic resonance image (MRI) showing multiple sites of bone marrow oedema on sacroiliac joint surface (**B**). “Bull’s head sign” on a chest CT (**C**).

**Figure 3 jpm-13-00516-f003:**
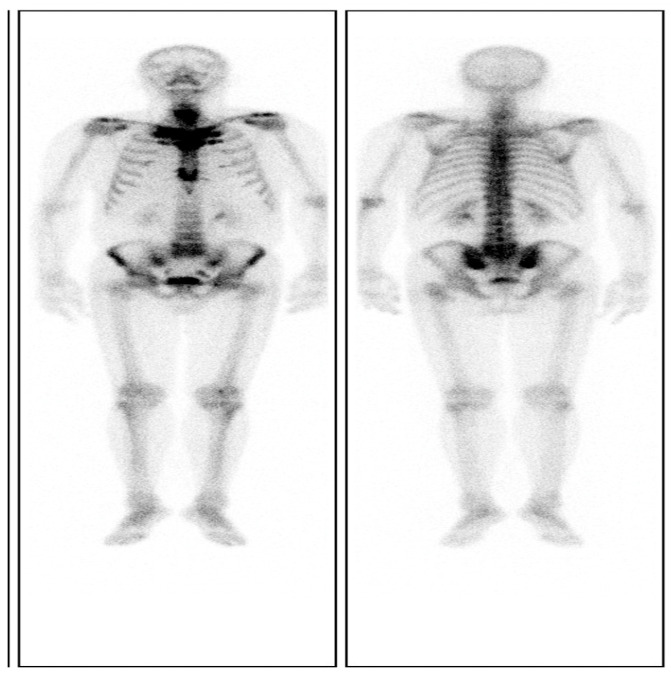
A ^99^Tc^m^-MDP SPECT whole-body bone scintigraphy revealing a remarkable accumulation in the cervical vertebra, sternoclavicular joint, bilateral first anterior rib, sternum, and sacroiliac joint due to inflammatory bone changes.

**Figure 4 jpm-13-00516-f004:**
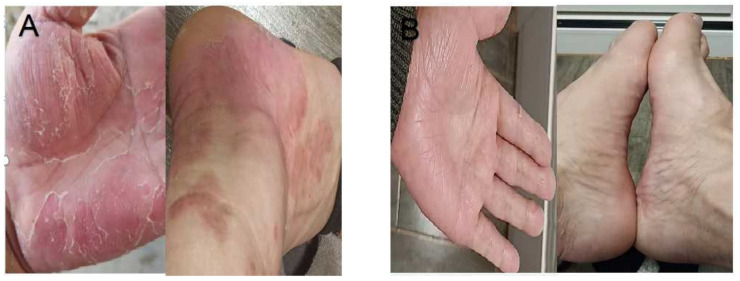
Psoriasis-like lesions were shown up on palms and feet after using adalimumab for 2 months (**A**). PPP on the patient’s palms and feet were almost in complete remission after using secukinumab for 2 months (**B**).

## Data Availability

Not applicable.

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
