# Peer review of "Successful Treatment of SAPHO Syndrome Complicated with Ankylosing Spondylitis by Secukinumab: A Case Report"

_jpm, 2023, doi:10.3390/jpm13030516_

Round 1

Reviewer 1 Report

Your case report piqued my interest. The article helps raise awareness of a rare disease related to psoriatic arthritis and axial spondylarthritis. However, it requires major revision to improve readability.

1 The discussion section should be rewritten. Some sentences appeared to repeat, and the wrong words were used.

2 In the abstract, the authors claimed that no previous cases of axial spondylarthritis were reported. However, they claimed in the discussion that one case of SAPHO with axial spondyloarthritis was treated with secukinumab. Please update the abstract to reflect that secukinumab was used to treat several Axial SpA cases associated with SAPHO.

Please include these references.

1Wendling D, Aubin F, Verhoeven F, Prati C. IL-23/Th17 targeted therapies in SAPHO syndrome. A case series. Joint Bone Spine. 2017 Dec;84(6):733-735. doi: 10.1016/j.jbspin.2017.05.016. 

 2Ji Q, Wang Q, Pan W, Hou Y, Wang X, Bian L, Wang Z. Exceptional response of skin symptoms to secukinumab treatment in a patient with SAPHO syndrome: Case report and literature review. Medicine (Baltimore). 2022 Aug 19;101(33):e30065. doi: 10.1097/MD.0000000000030065.

3 I recommend that authors review the literature for the articles relating to SAPHO and axial spondyloarthritis and elaborate on the clinical characteristics of these cases.

4 English grammar needs improving, particularly in the discussion. For example, line 98 ‘’notorious as a rare disease’’ line 100 ‘’it is a consensus that’’, line 101 ''an underestimated real incidence'', and line 104 ‘’middle-aged patients is more common with PPP’’

Author Response

Dear Reviewer:

Thanks for the comments on my manuscript and we had read the the suggestions carefully. According to the reviewers'opinion, the manuscript had been revised by correcting the errors including grammars, words, sentences, and all the errors pointed by the reviewers. Additionally, the references had been updated with all the articles provided. Moreover, we had provided more information in the discussion about the explanation of the case, such as the implication of “bull’s head sign” and the necessary image examinations for the diagnosis of SAPHO syndrome and AS. Meanwhile, the citations had been updated too, and the references provided by one reviewer had been added to suitable sentences to improve clarity and elucidation. Please see the attachment.

The discussion section should be rewritten. Some sentences appeared to repeat, and the wrong words were used.

Response: Thanks very much for the reviewer’s suggestions, which has helped us a lot. the discussion had been rewritten and updated by adding more information. The repeated information had been deleted, and the wrong words were corrected.

In the abstract, the authors claimed that no previous cases of axial spondylarthritis were reported. However, they claimed in the discussion that one case of SAPHO with axial spondyloarthritis was treated with secukinumab. Please update the abstract to reflect that secukinumab was used to treat several Axial SpA cases associated with SAPHO.

Response: Some articles reported secukinumab was used to treat SAPHO syndrome. We have added related references in discussion. We had search the PUBMED and did not find any study reporting the efficacy of secukinumab in treating SAPHO complicated with axial spondyloarthritis, with the keywords “SAPHO”, “axial spondyloarthritis” and “secukinumab”. However, it is the first time that we used secukinumab to treat SAPHO syndrome complicated with AS. We guessed that the reviewer could have gave rise a misinterpretation referring to this.

I recommend that authors review the literature for the articles relating to SAPHO and axial spondyloarthritis and elaborate on the clinical characteristics of these cases.

Response: the discussion had been updated with more information related to SAPHO and axial spondyloarthritis, with careful interpretation.

English grammar needs improving, particularly in the discussion. For example, line 98 ‘’notorious as a rare disease’’ line 100 ‘’it is a consensus that’’, line 101 ''an underestimated real incidence'', and line 104 ‘’middle-aged patients is more common with PPP’’

Response: the manuscript had been rewritten and proofread carefully to correct any possible error.

Reviewer 2 Report

This is a case of a patient with SAPHO syndrome associated with AS successfully treated with secukinumab. The report is interesting, but the manuscript should be improved:

Comments:

Line 15: “and has great improvement in outcome by optimizing management”: The sentence should be rewritten to improve clarity

 Line 19: Add: “was described” at the end of the sentence

 Line 21:  Secukinumab can significantly improve the patient’s cutaneous lesion and prevent musculoskeletal pain by substituting adalimumab. Change “can significantly improve” with “improved”

 Line 35: “aseptic” should be deleted before “osteoarthritis”

 Line 36: “attachment point inflammation”. Do you mean enthesitis?

 Line 41 “it is still lack of effective relieve management for SAPHO”. The sentence should be rewritten for clarity

 Line 41: Change “resolve” to “resolution”

 Line 51: “the sternum was shown tenderness pain”. The sentence should be rewritten to better clarity

 Line 69 “Notably, the characteristic “bull’s head sign,”. Please add a reference

 Line 70: “cheat”, please change to “chest”

 Line 71: SPECT use in this case report deserves a comment on why it was used and possible indications in SAPHO/AS.

 Line 80: ih. Do you mean subcutaneously (s.c.)?

 Line 84: “Once” may be deleted

 Line 87: Laboratory investigation results should be part of a new sentence since they are not related with adverse reaction.

 Line 100: “it is a consensus”. Better “there is consensus”

 Line 103: the sentence “but the middle-aged patients is more common with PPP accompanying cutaneous lesions, PPP is more common in, while young patients is usually presents as severe acne” is unclear an should be rewritten

Line 113: the term bacteriologic should be changed to bacterial

Line 114: add infection after Propionibacterium acnes  

Line 116: “as a prosperous area”: add: of research. The term prosperous might be changed with another one with a more scientific sound

Line 117: syndrome suffered from tonsillitis too, who presented. Better: suffering from tonsilitis presented…

Line 123: “Heretofore”. Change to another word

Line 125: Currently, there is no standard treatment for SAPHO syndrome: this phrase is a repetition. Should be deleted

Line 129: “curbing unscrupulous disease deterioration”. The sentence is unclear. Please rewrite it

 Line 170: we described in detail a case of SAPHO syndrome complicated by AS attained complete resolve after secukinumab substitution: The sentence is unclear. Please rewrite it

 Line 171: Change “prefer” to “preferred”

 Author Response

Dear Reviewer,

   Thanks very much for the suggestions, which has helped us a lot. Your suggestions included many errors concerning English writing. Therefore, we had scrutinized the manuscript carefully, and added more information to the discussion to make the description and explanation more clear. We have revised all the errors. Please see the attachment

 Comments:

Line 15: “and has great improvement in outcome by optimizing management”: The sentence should be rewritten to improve clarity

Answer: We have rewrite the sentence to improve clarity

Line 19: Add: “was described” at the end of the sentence

Answer: We have rewrite the sentence to improve clarity.

 Line 21:  Secukinumab can significantly improve the patient’s cutaneous lesion and prevent musculoskeletal pain by substituting adalimumab. Change “can significantly improve” with “improved”

Answer: We have changed the sentence.

 Line 35: “aseptic” should be deleted before “osteoarthritis”

Answer: We have deleted “aseptic” in the sentence.

 Line 36: “attachment point inflammation”. Do you mean enthesitis?

Answer: yes,we have replaced in the sentence..

 Line 41 “it is still lack of effective relieve management for SAPHO”. The sentence should be rewritten for clarity

Answer: We have rewrite the sentence to improve clarity.

 Line 41: Change “resolve” to “resolution”

Answer: We have changed the sentence.

 Line 51: “the sternum was shown tenderness pain”. The sentence should be rewritten to better clarity

Answer: We have changed the sentence.

 Line 69 “Notably, the characteristic “bull’s head sign,”. Please add a reference

Answer: We have added a reference to explain the sign.

 Line 70: “cheat”, please change to “chest”

Answer: We have changed the sentence.

 Line 71: SPECT use in this case report deserves a comment on why it was used and possible indications in SAPHO/AS.

Answer:We have added the in the discussion.

 Line 80: ih. Do you mean subcutaneously (s.c.)?

Answer:yes,we have changed in the manuscript.

 Line 84: “Once” may be deleted

Answer:we have changed in the manuscript.

 Line 87: Laboratory investigation results should be part of a new sentence since they are not related with adverse reaction.

Answer:we have changed in the manuscript.

 Line 100: “it is a consensus”. Better “there is consensus”.

Answer:yes,we have changed in the manuscript.

Line 103: the sentence “but the middle-aged patients is more common with PPP accompanying cutaneous lesions, PPP is more common in, while young patients is usually presents as severe acne” is unclear an should be rewritten.

Answer:we have changed in the manuscript.

Line 113: the term bacteriologic should be changed to bacterial

Answer:we have changed in the manuscript.

Line 114: add infection after Propionibacterium acnes  

Answer:we have changed in the manuscript.

Line 116: “as a prosperous area”: add: of research. The term prosperous might be changed with another one with a more scientific sound

Answer:we have changed in the manuscript.

Line 117: syndrome suffered from tonsillitis too, who presented. Better: suffering from tonsilitis presented…

Answer:we have changed in the manuscript.

Line 123: “Heretofore”. Change to another word

Answer:we have changed in the manuscript.

Line 125: Currently, there is no standard treatment for SAPHO syndrome: this phrase is a repetition. Should be deleted

Answer:we have changed in the manuscript.

Round 2

Reviewer 1 Report

I thank the authors for taking the advice on board and trying to improve the quality of the manuscript. However, there are still some issues that need sorting. The authors failed to incorporate all available research during the first revision. 

1 The authors reported this is the first reported case of secukinumab treating SAPHO syndrome complicated with AS.

This is not true. I suggest authors remove this sentence as Wang et al. demonstrated in their case series that four patients had axial spondyloarthritis (1 patient had bilateral sacroiliitis, and the rest had thoracolumbar inflammatory spinal lesions). Their bath ankylosing spondylitis disease activity index (BASDAI) was improved significantly following secukinumab therapy in patients with SAPHO.  

The findings reported in the case by Tu et al. that the SIJ lesions are not typical for sacroilitis what we see in axial spondyloarthritis patients (wormhole like bone destruction could be due to osteitis). Bone ankylosis is typical end result of osteitis in these patients. 

The evidence clearly suggests that Secukinumab was successfully used in patients separately for AS and AS and SAPHO together. Please include below reference and amend your manuscript accordiingly. 

Wang L, Sun B, Li C. Clinical and Radiological Remission of Osteoarticular and Cutaneous Lesions in SAPHO Patients Treated With Secukinumab: A Case Series. J Rheumatol. 2021 Jun;48(6):953-955. doi: 10.3899/jrheum.201260. Epub 2021 Mar 1. PMID: 33649072.

2 Teething grammatical errors are still present.

3 Never heard a word ''rough, narrow bone marrow oedema'' Please amend to just bone marrow oedema. 

Author Response

Dear Reviewer,

Thank you very much for the comments on my manuscript and we had read the your suggestions carefully. We have revised the manuscript carefully again.

1 The authors reported this is the first reported case of secukinumab treating SAPHO syndrome complicated with AS.

Response: We are very sorry for the error caused by our failure to search the reference. We read the article about case series of secukinumab treatment in SAPHO syndrome by Wang et al. We have revised the sentence in Abstract and added the reference in the article.

2 Teething grammatical errors are still present.

Response: We had scrutinized the manuscript carefully, and have revised some errors again.

3 Never heard a word ''rough, narrow bone marrow oedema'' Please amend to just bone marrow oedema. 

Response: We had amended the sentence.

Reviewer 2 Report

The authors fully answered my questions. 

Author Response

Dear Reviewer,

Thanks very much for the comments on my manuscript and we revised the manuscript accroding to the other reviewer. Please review it again. Thank you very much.

Best regards

Round 3

Reviewer 1 Report

Thank you for incorporating the changes to the manuscript.

Best Wishes,